# Broad-Spectrum Antivirals Derived from Natural Products

**DOI:** 10.3390/v15051100

**Published:** 2023-04-30

**Authors:** Wen-Jun Tian, Xiao-Jia Wang

**Affiliations:** Key Laboratory of Animal Epidemiology of the Ministry of Agriculture, China Agricultural University, Beijing 100193, China

**Keywords:** antiviral agents, host-targeting antivirals, direct-acting antivirals, natural products

## Abstract

Scientific advances have led to the development and production of numerous vaccines and antiviral drugs, but viruses, including re-emerging and emerging viruses, such as SARS-CoV-2, remain a major threat to human health. Many antiviral agents are rarely used in clinical treatment, however, because of their inefficacy and resistance. The toxicity of natural products may be lower, and some natural products have multiple targets, which means less resistance. Therefore, natural products may be an effective means to solve virus infection in the future. New techniques and ideas are currently being developed for the design and screening of antiviral drugs thanks to recent revelations about virus replication mechanisms and the advancement of molecular docking technology. This review will summarize recently discovered antiviral drugs, mechanisms of action, and screening and design strategies for novel antiviral agents.

## 1. Introduction

Viral infection has enormous social and economic impacts [1], but there are as yet only a few drugs available to treat viral infections. Furthermore, treatment is becoming more difficult because of virus mutation [2]. Different strategies are required to screen for and design antiviral drugs to counter viral infections. Host-targeting antivirals (HTAs) and direct-acting antivirals (DAAs) are promising approaches. DAAs target the virus itself, including structural proteins, non-structural proteins, and nucleotides, with specificity but a high likelihood of resistance. DAAs include polymerase inhibitors, protease inhibitors, inhibitors of transcriptase, and reverse transcriptase. HTAs target host factors indispensable for viral replication, with broad-spectrum activity and a low likelihood of resistance. Here, we focus on natural antiviral agents for three kinds of viruses, coronavirus, influenza virus, and herpes virus, which are widespread and difficult to contain.

A virus is a microorganism that is strictly dependent on the host body for reproduction. Viruses carry either ribonucleic acid (RNA) or deoxyribonucleic acid (DNA) as their genetic material, which can be single-stranded or double-stranded; the two structures represent different replication processes. Coronavirus infection is famous for the severe acute respiratory syndrome caused by severe acute respiratory syndrome coronavirus (SARS-CoV) in 2003 [3]. A decade later, Middle East respiratory syndrome coronavirus (MERS-CoV), the second human coronavirus, emerged in 2012 in Saudi Arabia. Unfortunately, a third novel coronavirus, SARS-CoV-2, caused a global pandemic in early 2020. This virus caused COVID-19 [4]. The coronavirus pandemic not only threatened health and life but also affected world economic development and social stability.

Influenza viruses are classified into A, B, C, and D types, based on their antigenicity [5]. The surface glycoproteins of hemagglutinin (HA) and neuraminidase (NA) are the main characteristics of the influenza virus. Influenza A (IAV) has 18 different HA subtypes (H1–H18) and 11 different NA subtypes (N1–N11) [6]. Influenza A viruses have a wide variety of hosts, with birds as their main natural host and pigs as a reservoir through which influenza viruses are transmitted to humans [7]. The most severe recent influenza pandemic is the Spanish flu of 1918, which killed more than 50 million people worldwide [8]. In 2009, the H1N1 flu pandemic killed an estimated 105,700 to 395,600 people worldwide [9].

The main characteristic of the herpes simplex virus (HSV) is a latent infection, which can be reactivated from this latent state, giving rise to a recurrence of the disease. Symptoms caused by HSV infection mainly include oral and genital herpes, both of which are transmitted by contact [10]. HSV has a very high occurrence rate worldwide, and according to previous reports, in 2016, an estimated 491.5 million people were living with HSV-2 infection, and an estimated 3752.0 million people had HSV-1 infection [11]. HSV infection seriously affects quality of life and health, and because of latent infection, the effects may last a long time.

In addition to influenza viruses, which emerge every year, and HSV, which affects many adults, there is also a need to focus on coronaviruses, which may break through the interspecies barrier to infect people. Beyond developing safer and more effective vaccines, we also need to discover more antiviral drugs that can be used for treatment. Only by controlling the spread of a virus in these two ways can we better protect humans. Our laboratory focuses on the screening and mechanism studies of antiviral drugs. This review describes the pathogenic mechanism of some human viruses and the existing inhibitors in an effort to contribute to the on-going development of antiviral drugs.

## 2. Life Cycles of Different Viruses

### 2.1. Life Cycles of Coronavirus

Coronavirus (CoV) is an enveloped virus belonging to the Coronaviridae family, Nidovirales order, with a 24.5–31.8 kb positive single-stranded RNA genome, the largest RNA virus genome [12,13]. The CoV spike protein binds to a specific receptor on the cell surface [14,15,16]. This process causes structural changes in the cell membrane, so the virus can enter the cell via membrane fusion or endocytosis [17]. Viral genes then enter the cytoplasm and hijack the cytokines for their own life cycle. The CoV genome encodes four structural proteins: spike, envelope, membrane, and nucleocapsid protein. CoV non-structural proteins are complex, encoded by two large open reading frames, and require multiple protease modifications. The function of non-structural proteins, including Nsp12 (RNA-dependent RNA polymerase, RdRp), 3CLpro (Mpro), and PLpro, is indispensable to CoV replication [12]. The CoV genome uses cap-dependent mechanisms to initiate translation. In this process, a viral protein such as a nucleocapsid protein hijacks the host eIF complex to promote viral protein translation, while regulating the translation of some host proteins [18] (Figure 1).

### 2.2. Life Cycles of Influenza Virus

The influenza virus is an enveloped virus belonging to the Orthomyxoviridae family, with a genome consisting of 8 segments, single negative-strand RNAs [19]. Unlike CoV, the influenza virus uses its HA protein to bind to a sialic acid receptor on the cell surface [20]. Similar to CoV, however, the subsequent entry process of the influenza virus consists of 3 steps: attachment of the viral hemagglutinin to the sialic acid receptors, membrane fusion or endocytosis, and M2-mediated uncoating [21]. NA is an enzyme that enables the virus to be released from the host cell. Notably, the matrix protein 2 (M2) ion channel protein of the virus mediates the fusion of the viral envelope with the endosomal membrane in a low pH-dependent manner so that the viral genome is released into the cytoplasm [22,23], while the coronavirus process depends on a cell protein (Figure 1).

The transcription mechanism of the influenza virus is quite complicated. After the viral ribonucleoprotein (vRNP) complex, consisting of RNAs, nuclear protein, and RdRp, invades the cytoplasm, the whole complex enters the nucleus, mediated by nuclear localization sequences (NLS) carried by vRNP, where the virus completes transcription [24,25]. The viral polymerase captures the 5′ cap of nascent host-capped RNAs, which is called cap-snatching [26]. The progeny vRNPs are exported to the cytoplasm mediated by the non-structural protein NS2, located on the membrane, to assemble the progeny virions [27]. Virions are released from the cell through NA cleavage for a new round of transmission, which means that the binding of HA and the cleavage of NA must reach a dynamic equilibrium state to maximize transmission efficiency [28].

### 2.3. Life Cycles of the Herpes Simplex Virus

The herpes simplex virus is an enveloped, double-stranded DNA virus belonging to the Herpesviridae family. Its genome is about 152 kb and encodes more than 80 different open reading frames [29]. As a DNA virus, HSV is quite different from CoV and influenza viruses. The entry process and replication cycle of HSV are more complicated. HSV has not one but five surface glycoproteins, including gC, gB, gH, gL, and gD. HSV entry begins with gB or gC binding to HS proteoglycans (HSPG) on the cell surface. Then, gD promotes membrane fusion through interaction with the gH/gL heterodimers. In addition to HSPG, the cell receptors of HSV also include the herpesvirus entry mediator (HVEM), nectin-1, nectin-2, 3-O-heparan sulfate (3-OS-HS), immunoglobulin-like receptor type 2 (PILRα), non-muscular myosin heavy chain IIA (NMHC-IIA), and myelin-associated glycoprotein (MAG) [30]. Viral DNA synthesis and nucleocapsid assembly take place in the nucleus, while virion processing and maturation take place in the cytoplasm [31], which means that the HSV replication cycle requires nucleocytoplasmic transport.

The virus life cycle, in addition to viral proteins, also involves the regulation of a variety of cytokines. All of these involved proteins can serve as potential sites for antiviral drug action.

## 3. Therapeutic Natural Antivirals for Virus Entry

### 3.1. Natural Products Targeting Coronavirus Entry

#### 3.1.1. Receptors

CoV invasion is triggered by S protein binding with a receptor on the cell surface. The first therapeutic strategy is to block the receptors with antibodies or drugs. Several peptides are designed to target the receptor. MERS-5HB contains five copies of the S protein N-terminal, which can inhibit MERS-CoV infection by mimicking the binding of the S protein to its receptor dipeptidyl peptidase-4 (DPP4) [32]. Similarly, the T20 peptide, designed to inhibit HIV entry [33], also mimics the CoV fusion peptide, preventing virus entry [34]. The problem with this strategy is that peptides and antibodies are proteins, and their instability affects bioavailability [35], which requires modification for clinical use [36,37]. Furthermore, CoVs use specific receptors for entry, which gives the antiviral high specificity but also makes it vulnerable to the effects of virus mutation, leading to resistance [38]. Natural products, rather than compounds, may be a good choice for these structurally diverse transmembrane proteins since some natural products can attack multiple targets.

#### 3.1.2. Membrane Protease

As mentioned above, the CoV S protein binds to a specific receptor to initiate virus entry. After binding, transmembrane protease serine (TMPRSS) is activated to cleave the S protein, dividing it into S1 and S2, which is necessary for subsequent membrane fusion. In addition, TMPRSS activates the inflammatory reaction by regulating the NF-κB pathway. TMPRSS may, therefore, serve as a potential target where an antiviral could effectively inhibit viral entry. According to at least one report, inhibition of TMPRSS2 with camostat can reduce MERS-CoV and SARS-CoV production in vitro. Unfortunately, it may not effectively treat patients with COVID-19 [38]. Propolis is produced by honeybees from plant exudates and is used in traditional herbal medicine. Studies indicate that propolis reduces TMPRSS2 expression and that propolis may be used as a supplement (rather than a therapeutic drug). It has been reported that alpha-1-antitrypsin (AAT) may antagonize infection induced by SARS-CoV-2 [39]. AAT is a serine protease inhibitor (SERPIN) proven to inhibit TMPRSS-2. In recent years, some researchers have used molecular simulation and high-throughput screening methods to screen potential natural inhibitors of TMPRSS, including phyllaemblicin G7, neoandrographolide, and kouitchenside I [40], among others (Figure 2). Although the inhibitory effect of these natural products on A has not been verified, it provides a new direction for research and development of new drugs to solve CoV infection.

Herpes virus entry is quite different from that of an RNA virus. Herpes virus entry is facilitated by membrane protease activity, and the transport of viral capsids to the nuclear periphery is necessary for herpes virus entry [41]. Lee et al. reported that a proteasome inhibitor bortezomib, which is an FDA-approved cancer drug, inhibits entry of a variety of herpes viruses, including HSV-2, pseudorabies virus (PRV), and bovine herpesvirus 1 (BoHV-1), by blocking the cell membrane protease [42].

### 3.2. Natural Products Targeting Membrane Fusion and Endocytosis

After binding, the virus enters the cytoplasm via membrane fusion or the endocytosis pathway. Most envelope viruses enter cells by these pathways, which involve cytoskeletal changes and lysosome acidification. Inhibitors targeting this process tend to have a broad spectrum of inhibition.

Zhao et al. demonstrated a broad inhibitor, frog-defensin-derived basic peptide (FBP), which can block low pH-induced HA-mediated fusion and antagonize endosomal acidification against virus infection. Moreover, FBP can also block coronavirus spike-mediated cell–cell fusion in 293T/ACE2 cell endocytosis [41]. Lysosome acidification depends on the exchange of metal ions, such as hydrogen, or calcium. Reports have shown that the Ca^2+^ channel as a host cell surface receptor, mediates the entry of IAV [42,43]. Indeed, the selective Ca^2+^ channel inhibitor nisoldipine has been shown to inhibit the influenza A virus by impairing the internalization of the influenza virus into host cells [44] (Figure 2).

Previous studies have suggested that in this process, endosomal acidification plays an important part [15,45,46], by which CoVs release their genomes into the cytoplasm. Chloroquine (CQ), a 9-aminoquinoline that can increase the pH of acidic vesicles [47], has been widely used to treat human diseases, such as malaria and amoebiasis [48]. Vincent proved that chloroquine effectively inhibits SARS-CoV infection [47], and Wang et al. also indicated that chloroquine inhibits SARS-CoV-2 infection in vitro [49]. In addition, it has been reported that the bovine herpes virus BoHV-1 utilizes a low-pH-mediated endocytosis mechanism to invade cells [50], which suggests that CQ could be used as a herpes simplex virus inhibitor. Because chloroquine also regulates immune function [48], the mechanism of chloroquine against SARS-CoV-2 is still unclear. Unfortunately, in recent years, some reports point out that CQ is useless in the clinical treatment of COVID-19 [51]. Therefore, more clinical data is required before widespread treatment.

Previous reports suggest that HSV-1 entry induces remodeling of the cytoskeleton and regulates the activation of cofilin [52]. Our research demonstrated that the small molecule compound RAF265 inhibits a variety of viruses, including HSV and PEDV, by mediating cytoskeleton rearrangement [53]. In addition, it has been reported that 17-AAG and AT533, which are Hsp90 inhibitors, inhibit HSV infection by blocking HSV attachment and inducing F-Actin Assembly [54].

## 4. Natural Products Targeting Viral Translation

All viruses need to hijack the cell translation system to perform viral protein translation after their genome is released into the cytoplasm. The eukaryotic translation mechanism consists of three steps: initiation, elongation, and termination [55,56]. Eukaryotic initiation factors (eIFs) play an important role in translation initiation, which is a rate-limiting step [57,58,59]. The initiation beginning with eIF4F, consisting of eIF4A, eIF4E, and eIF4G, binds to the m^7^GTP residue at the 5′- untranslated region (5′UTR) of mRNA for cap-dependent translation [60,61]. Phosphorylation of eIFs is an important part of regulating translation [62] (Figure 2).

### 4.1. Targeting the Translation Initiation Factor

#### 4.1.1. eIF4A

The eIF4A factor is a DEAD-box helicase that binds and unwinds 5′UTR structures of mRNA [63]. Based on the CoV translation mechanism, eIF4A inhibitors may block viral replication. Silvestrol, a natural compound isolated from the plant *Aglaia*, used in traditional Chinese medicine, also inhibits virus replication by blocking eIF4A [64,65].

#### 4.1.2. Phosphorylation of eIF4E (p-eIF4E) and eIF4E

Similar to eukaryotic translation regulation, the CoV mRNA 5′ cap binds eIF4E to control eIF4F complex formation and interferes with cellular translation via phosphorylation on serine 209 [66]. Similarly, the regulatory role of p-eIF4E in HSV translation was reported in the early 2000s [67]. In addition, unlike eIF4E, which is an essential protein for all cells [68,69], p-eIF4E occurs only in tumor cells or virus-infected cells. Homoharringtonine (HHT) is a natural product well known in China to resolve chronic myeloid leukemia (CML) and some kinds of tumors. Our research team found in previous experiments that HHT presents effective antiviral activity, which antagonizes the phosphorylation level of eIF4E [70]. We found that HHT inhibited the translation level of a variety of viruses, including HSV and CoV, and restricted viral infection. In subsequent experiments, we found that p-eIF4E was involved in regulating some cytokines that affect PEDV replication. Therefore, we believe that p-eIF4E is a potential antiviral target [66].

Since viruses and cells use a common translation system, inhibitors targeting this system are difficult to apply in a clinical setting due to their cytotoxicity. However, the regulatory factor p-eIF4E exists only in abnormal cells, so the inhibitors targeting it may be less cytotoxic.

## 5. Natural Products Targeting an Innate Immune Response

The innate immune response is the most important part of cell resistance to viral infection. This process is triggered by pattern recognition receptors (PRPs), including the Toll-like receptor (TLR) family, which recognize pathogen-associated molecular patterns (PAMPs), activating a series of responses, including nucleocytoplasmic transport of some proteins or complexes and activation of some cell receptors [71]. Briefly, after recognition, retinoic acid-inducible gene I (RIG-I) or melanoma differentiation gene 5 (MDA5) will be activated, which binds to the mitochondrial adapter protein MAVS/IPS-1, and recruits TNF receptor-associated factor (TRAF). TRAF activates IκB kinase (IKK)-related kinases, such as TANK-binding kinase 1 (TBK1), which phosphorylate Interferon Regulatory Factors 3 and 7 (IRF3/IRF7) [72,73,74,75]. Phosphorylated IRF3 forms a homodimerization and enters the nucleus, where it becomes a complex with transcription co-activator CREB (cAMP responsive element binding)-binding protein (CBP)/p300 [76,77,78]. Finally, IFN molecules bind to the cell receptors and trigger activation of Janus kinase–signal transducers and activators of the transcription (JAK–STAT) signaling cascade to form the IFN-stimulated gene factor 3 (ISGF3), which will induce expression of hundreds of antiviral genes and establishment of an antiviral state [79,80,81]. This is a complicated process that involves many cellular proteins, which means one or more sites can be targeted for different antivirals, or antivirals with multiple targets (Figure 2).

Vitamin C is an essential nutrient deficiency that results in scurvy. Individuals with scurvy are highly susceptible to fatal infections, such as pneumonia [82]. Clinical trials have shown that high doses of Vitamin C have effects against virus infection [82,83,84,85], and research suggests that the antiviral effect might be attributable to the production of antiviral cytokines (IFNs), free radical formation, or direct binding to the virus [86]. In addition, vitamin D has shown regulatory effects on both innate immunity and acquired immunity [87]. Indeed, both vitamins have been reported to be effective in treating COVID-19 by regulating the body’s immune system [86,88,89].

CQ, as mentioned above, also has an immunomodulatory ability [90] downregulating the expression of TLRs (Toll-like receptors) and TLR-mediated signal transduction, and decreasing the release of TNF-α, IL-1, and IL-6, caused by SARS-CoV-2. Furthermore, CQ inhibits the interactions between TLRs and viral RNA, and regulates the production of IFNs [91].

Some immunomodulators derived from natural products that treat tumors may be repurposed to regulate the immune response to viral infections. Curcumin has been shown to suppress breast cancer cell viability by downregulating expression levels of TLR4 and IRF3, which then inhibits expression of type I IFN (IFN-α/β) [92]. Researchers reported that ω-3 polyunsaturated fatty acids (PUFAs), abundant in fish, suppress inflammatory cytokine storms in patients, induced by TLRs and NF-κB [93]. Immune factors are highly conserved, so some drugs that regulate the immune response may be repurposed as drugs to restrict viral infections.

In recent years, to address the COVID-19 pandemic, researchers have accelerated the screening process for antiviral drugs. They found that Quercetin 3-O-(6″-galloyl)-beta-D-galactopyranoside, Tribuloside, and Rutin may be potential natural inhibitors for TMPRSS by molecular simulation and high-throughput screening methods. In addition, researchers have found that the natural products, including phyllaemblicin G7, neoandrographolide, and kouitchenside I, have the structural basis to inhibit the interaction of IRF3 with SARS-CoV-2 [94].

## 6. Virus-Targeting Antivirals

### 6.1. Targeting Coronavirus Protein

As mentioned above, CoV encodes five structural proteins. N protein forms nucleocapsid, which participates in viral genome transcription and protein translation. The M protein determines the shape of the viral envelope and drives the formation of the viral envelope. The E protein may be involved in virus assembly and budding. In addition, CoV encodes 16 non-structural proteins, most of them key proteins in viral replication and replication regulation, such as Nsp3 (papain-like protein), Nsp5 (3CLpro), and Nsp12 (RdRp). Nsp1 is involved in the regulation of the innate immune response [95]. Nsp4 and Nsp6 induce membrane rearrangement to construct a double-membrane vesicle (DMV), in which the viral replication transcription complex (RTC) is anchored [96]. Nsp12, the core of RNA-dependent RNA polymerase (RdRp), works together with Nsp7/8 to transcript and replicate [97]. The activation of innate immunity requires both the stimulation of pathogenic microorganisms, and cellular immune response. Multiple viral proteins of CoV, such as Nsp1, Nsp3, and Nsp5, can inhibit host innate immune protection against normal replication.

In addition to their role in hijacking cellular proteins, viral proteins are equally important to the virus’ own life cycle. Unfortunately, drugs targeting viral proteins require highly specific design and screening processes. Although they may ultimately face drug resistance, they should have a good therapeutic effect in the short term.

#### 6.1.1. Nsp1

Nsp1 is unique to alpha- and beta-CoV and has been found to inhibit interferon response in PEDV and SARS-CoV-2 by promoting the degradation of the CREB-binding protein (CBP) [98]. PEDV Nsp1 also suppresses NF-κB activity against the expression of IFNB. In addition, SARS-CoV-2 Nsp1 has a conserved function in inhibiting host protein synthesis [99]. Thus, Nsp1 appears to be a potential target for therapeutics. As of now, there is no effective drug that can inhibit the function of Nsp1. Fortunately, with the structure of Nsp1 revealed, researchers have found some natural products with potential inhibitory activity against Nsp1 through molecular docking. It has been reported that amontelukast sodium hydrate may be the potential inhibitor for SARS-CoV-2 Nsp1 [100]. Additionally, the researchers found that Golvatinib, Gliquidone, and Dihydroergotamine bind to Nsp1 and have a potential inhibitory effect against SARS-CoV-2 [101].

#### 6.1.2. Nsp3 (PLpro) and Nsp5 (3CLpro)

As proteases, Nsp3 (PLpro) and Nsp5 (3CLpro) are mainly responsible for cutting and modifying other non-structural proteins. These non-structural proteins have biological functions only after being cleaved and modified. Therefore, both Nsp3 and Nsp5 have been considered potential targets for antiviral agents. Hesperidin, a flavonoid, can inhibit the cleavage activity of 3CLpro in a dose-dependent manner [102]. Quercetin is also a flavonoid and has been reported as an inhibitor of 3CLpro. Indeed, a variety of flavonoids have been reported as inhibitors of Nsp3 and also have immune regulatory effects [103]. In addition, the researchers found that tea catechins may be potential drug candidates for CoV infection by inhibiting 3CLpro [104].

#### 6.1.3. RdRp

RNA virus genomic RNA can be used with RdRp as a template to synthesize progeny RNA. Therefore, RdRp is an important target for antiviral agents. Murphy et al., for example, reported that the nucleoside analogue GS-441524 inhibits feline coronavirus replication by blocking the binding of RdRp with viral RNA [105]. Subsequent reports have shown that GS-441524 has low toxicity [106] to cats and can be an effective treatment for FIP [107]. Similarly, remdesivir has the same skeleton as GS-441524, which has been approved as an effective treatment for COVID-19 infection, but some of its side effects are undesirable [108,109]. Furthermore, remdesivir resistance has been reported, which reduces its clinical usefulness [110,111].

Interestingly, the amentoflavone, a bioflavonoid, seems to bind with a variety of non-structural proteins of CoV, including Nsp3, Nsp5, Nsp12, and Nsp15 [112]. It has been shown that flavonoids basically inhibit viral protease; therefore, flavonoids may be in the drug library for antiviral drug screening in the future.

### 6.2. Targeting the Influenza Virus Protein

Influenza virus proteins have many functions, from ion channels (M2) to nuclear localization signals (NS2), and proteases. RNA synthesis of the influenza virus is accomplished by multiple RNA polymerases [113]. As mentioned above, IAV transcription takes place in the nucleus, and polymerase basic 1 (PB1) is the most important RdRp for IAV. Some viral proteins are also involved in regulating the innate immune response, including PB1-F2 and NS1. Hemagglutinins (HA) and neuraminidase (NA) also play important roles in virus transmission. These key viral proteins are also potential targets for antiviral drugs.

#### 6.2.1. HA and NA

The glycoproteins HA and NA play important roles in virus entry and release, which are the first targets for restricting viral infection. Indeed, previous specific drugs against the influenza virus include NA inhibitors, such as zanamivir and oseltamivir, which inhibit the cleavage of NA and limit the release of influenza viruses. However, resistance to zanamivir and oseltamivir develops rapidly in clinical settings because of virus variation [114,115]. Similar to NA, small molecule inhibitors targeting HA, such as Arbidol (a fusion inhibitor), are vulnerable to the same resistance. Newer inhibitors are therefore needed. Oleanolic acid (OA) has been shown to be a mild influenza hemagglutinin (HA) inhibitor [116]. In a follow-up experiment, OA derivatives were synthesized and showed good inhibitory activity against influenza viruses in vitro [117]. It has been reported that punicalagin inhibits agglutination of chicken RBCs caused by viruses, which can be used with oseltamivir against IAV [118]. Curcumin, a regulator of TLR4 and IRF3, as mentioned above, has been reported to inhibit HA function and also affect envelope integrity to deal with IAV infection [119]. It is hypothesized that curcumin can treat viral infection in a variety of ways, or by targeting multiple targets, and further verification is required.

#### 6.2.2. RNA Polymerase

The diversity and high mutation rate of HA and NA make them prone to drug resistance, but the conservation of RNA polymerase may be a target to solve the problem of drug resistance. The influenza virus polymerase system, which includes PB1, performs RNA synthesis; PB2 recognizes cap structures, and PA-PB1 interactions mediate polymerase entry into the nucleus. Therefore, the RNA polymerase of the influenza virus is a potential target for exploring antiviral agents.

To inhibit polymerase activity, it is a good strategy to block the interaction between PA and PB1. Mélin, et al. reported that the lead compound LM146 displays a strong affinity with PB1 [120]. Huang et al. suggest that roscovitine, a cyclin-dependent kinase inhibitor (CDKI), can be used as a PB2 cap inhibitor against influenza A virus infection [121]. The compound pimodivir, which is a PB2 cap inhibitor, has entered the third phase of clinical trials. However, with the mutation of viral proteins, pimodivir has shown some resistance [122,123]. In addition to blocking the interaction between PB1 and PA, PA protein also represents an attractive target for new antiviral drugs because it is highly conserved. Researchers have found some natural products, such as tea catechin, which have been shown to inhibit the influenza virus in vitro [124]. It has also been reported that a series of 1, 3-cis-2-substituted-1-(3, 4-dihydroxybenzyl)-6, 7-dihydroxy-1, 2, 3, 4-tetrahydroisoquinoline-3-carboxylic acid derivatives are potent PA inhibitors that can inhibit influenza virus replication in vitro [125].

Chemical compound DAAs easily lose their inhibitory effect with the mutation of viral proteins; thus, it is necessary to constantly screen new drugs or search for natural products that target multiple sites, which are thus less vulnerable to resistance.

### 6.3. Targeting the Herpes Simplex Virus Protein

HSV is a large DNA virus that encodes a large number of viral proteins with different functions. The most important viral proteins include origin-binding protein UL9, single-stranded DNA binding protein ICP8, DNA polymerase complex UL30/42 [126], and the mediate-early viral protein ICP0 [127].

#### 6.3.1. Surface Glycoproteins

HSV invades cells by using surface glycoproteins, which are important targets for screening the antiviral agents. Majmudar et al. found that sulfated pentagalloylglucoside (SPGG) may be an effective inhibitor of HSV-1 entry by interacting with HSV-1 glycoprotein [128]. In addition, Wu et al. reported that the natural compound 1-(1-benzofuran-2-yl)-2-[(5Z)-2H,6H,7H,8H- [1,3] dioxolo [4,5-g] isoquinoline-5- ylidene] ethenone, which was screened by the molecular docking method, can interact with HSV surface glycoprotein gD, and shows an inhibitory effect for HSV [129]. In addition, some peptides also show the ability to bind the HSV glycoprotein in vitro, such as peptides U-1 and U-2 [130], although the effect of these peptides is relatively weak.

#### 6.3.2. DNA Polymerase

The classic HSV inhibitor is acyclovir, which is the basis of 2′-deoxiguanosin, targeting HSV DNA polymerase to restrict viral DNA replication. It has been reported that psoromic acid, a bioactive lichen-derived compound, could act against HSV-1 and HSV-2 by blocking HSV DNA polymerase [131]. Wald et al. reported that pritelivir, an inhibitor of the viral helicase–primase complex, countered HSV infection both in vivo and in vitro [132].

Different kinds of viruses have different kinds of viral proteins. Antiviral drugs that target individual viral proteins are more specific, which means lower cytotoxicity and better therapeutic efficacy. Different viruses invade cells through different receptors. Based on this, antiviral drugs have high specificity and can effectively inhibit viral entry. After entering, all viruses need to release their genomes into the cytoplasm, and then use the host system for their own replication. In this process, the translation of viral proteins needs to hijack the host translation system. In addition, different viruses induce innate immunity in different ways, but they use the cellular proteins that are used in immune regulation. These are common to different viruses, which means they are potential targets for broad-spectrum antiviral drugs. Therefore, antiviral agents should be screened and used specificity to make them more effective.

## 7. Discussion

Developers of antiviral drugs face a dilemma. On the one hand, DAAs show resistance due to viral mutations, and on the other hand, HTAs have greater cytotoxicity. Our review considers natural products targeting both viral proteins and cellular proteins, which is one of the characteristics of natural products. CQ, for example, not only inhibits viruses that are dependent on low pH, but also regulates the immune system. Some natural products of bases or acids can kill the virus directly. Other purified natural products are compounds with a well-defined structure that have the structural basis of targeted drugs. Natural products offer a potentially rich field for future drug screening.

High-throughput screening (HTS) methods have become an important tool for discovering antiviral drugs. Unfortunately, HTS is very costly and time consuming; it requires the development of specific assays for selected pharmaceutical targets, and a compound library must first be synthesized. With the development of virtual screening (VS) methods, the fast and cheap identification of lead and hit structures of target proteins is possible. Swamy et al. described how available antiviral candidates can be reused as a potential SARS-CoV-2 3CLpro inhibitor, using molecular docking studies. They found that some types of compounds from the antiviral library, such as Mitoxantrone and Leucovorin, are potential inhibitors of SARS-CoV-2. In addition, glutamic acid (Glu166) of 3CLpro is a key residue holding and forming a stable complex of these compounds by forming hydrogen bonds and salt bridge [133]. In addition, the mechanism of the antiviral action of glycyrrhizinic acid (GA), a natural triterpene saponin, has been elucidated by docking and molecular dynamics studies with GA molecules with the 3CLpro of SARS-CoV-2 [134]. However, the mechanism of GA is still controversial.

This screening technique could break through the limits of the drug library to screen for more natural compounds, which offers the possibility of high resistance to viral mutation and lower cytotoxicity. In subsequent studies, this screening technology can be combined with biological experiments and clinical trials, and safer and more effective antiviral drugs may be screened. In addition, by better understanding the mechanism of virus-host cell interaction, natural products from the library can be found more efficiently that share structural similarities with antiviral compounds, thereby reducing toxicity and expense.

## Figures and Tables

**Figure 1 viruses-15-01100-f001:**
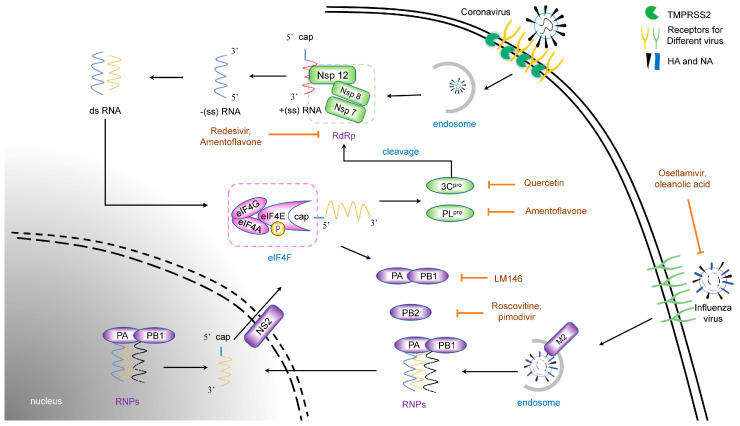
The inhibitors targeting viral proteins. CoV and IFV bind the different receptors and use different membrane proteases, meaning the different inhibitors for binding. Their entry into cells both relies on the endosomal pathway mediated by a low pH, and the same inhibitor chloroquine (CQ) may work. The RNA virus replication dependent on their RNA-dependent RNA polymerase (RdRp), which is highly conserved among different strains, means the potential target for antiviral agents. Black arrows indicate signaling pathways, orange arrows indicate inhibition.

**Figure 2 viruses-15-01100-f002:**
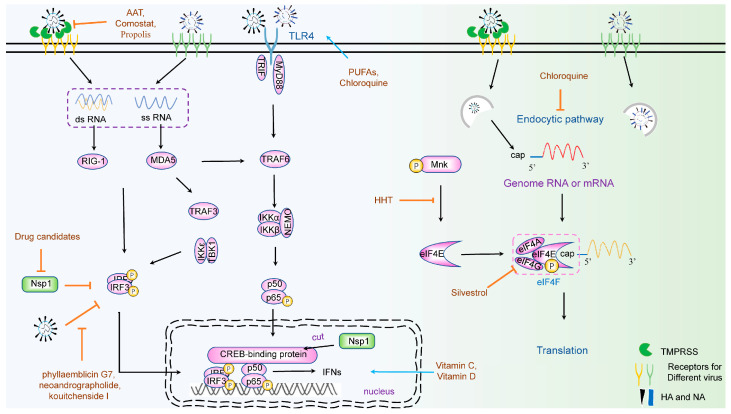
The inhibitors targeting host factors. Initial immune response induced by a virus and also regulated by a virus. Immune-regulating drugs need to balance the immune system, such as Vitamin C and Vitamin D. In addition, some drugs need to stimulate the immune response to produce antiviral factors, such as PUFAs and Chloroquine. The eukaryotic translation system of host cells is essential for viral translation. CoV and IFV hijack the host eIF factors to trigger cap-dependent translation to produce viral proteins. Therefore, translation inhibitors with less toxic side effects can be used as potential antiviral drugs, such as HHT. Black arrows indicate signaling pathways, orange arrows indicate inhibition, and blue arrows indicate activation.

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
