# Peer review of "Broad-Spectrum Antivirals Derived from Natural Products"

_viruses, 2023, doi:10.3390/v15051100_

Round 1

Reviewer 1 Report

The manuscript describes the potential antiviral therapy against three kinds of viruses: coronavirus, influenza virus, and herpes virus. The Authors aimed to summarize recently discovered antiviral drugs, their mechanisms of action, screening, and design strategies for novel antiviral agents. As the title of the manuscript suggests the Authors aimed to focus on natural antiviral products.

In my opinion, the paper in its present form cannot be recommended for publication – major revision is required.

The first and the second paragraph are well described but the next paragraph describing antiviral agents against the herpes virus is insufficient. In my opinion, the Authors should rather cancel herpes virus from the manuscript and more concentrate on the influenza virus and coronavirus.

The manuscript is difficult to read because the very general information is mixed with scientific details. Also, nonessential data such as flavivirus genome or feline coronavirus were described.

The Authors should decide if they focus on natural antiviral products or rather they compare natural products with antiviral drugs. In the manuscript the description of drugs dominates, and only a few natural compounds were mentioned. The title and the abstract should correspond to the main text but in this version of the manuscript it is not.

 The references in the text should be improved, sometimes author’s name and year are included instead number.

Also, unnecessary abbreviations were in the text, for example, psoromic acid PA) – this term is the only one in the text.

Author Response

Manuscript ID: viruses-2303505

Title: Broad-spectrum antivirals derived from natural products

Responses to reviewer1

Reply: Thank you for your constructive comments and suggestions. Considering the suggestions, we have conducted a thorough revision to make the study clearer and stronger. All changes are highlighted in blue in the revised version, a "Marked Up Manuscript File," and responses to the reviewers' points are addressed in detail below.

1、The manuscript describes the potential antiviral therapy against three kinds of viruses: coronavirus, influenza virus, and herpes virus. The Authors aimed to summarize recently discovered antiviral drugs, their mechanisms of action, screening, and design strategies for novel antiviral agents. As the title of the manuscript suggests the Authors aimed to focus on natural antiviral products. In my opinion, the paper in its present form cannot be recommended for publication – major revision is required.

Response: We wish to thank the reviewer for your instructive comments on the last version of this manuscript. We have revised the manuscript and addressed your concerns, as stated in the updated manuscript. All changes are highlighted in the revised version, a "Marked Up Manuscript File," and responses to the reviewers' points are addressed in detail below.

2、The first and the second paragraph are well described but the next paragraph describing antiviral agents against the herpes virus is insufficient. In my opinion, the Authors should rather cancel herpes virus from the manuscript and more concentrate on the influenza virus and coronavirus.

Response: We appreciate the reviewer's comment. We have made this change in the revised paper, in which we added more details about antiviral agents against herpes virus, a DNA virus, to make the study integrally. Both DAN and RNA viruses utilize host cellular translation system and mediate similar immune response. It is the reason that we commentated the antiviral agents for three viruses in Sections 3.3,4 and 5, and separately described in Sections 3.1, 3.2 and 6 according to their difference. We summarized the differences and similarities among different types of viruses and suggested the antiviral agents will be screened and designed specifically and effectively. Please see page 5, lines 22-25., and page 4, line 43-48 (unmarked version).

3、The manuscript is difficult to read because the very general information is mixed with scientific details. Also, nonessential data such as flavivirus genome or feline coronavirus were described.

Response: We appreciate the reviewer's comment. We included an advances in viral replication cycle before introducing antivirals, to better understand the specific antiviral target. A general information you mentioned shouldn't be included in, we have therefore deleted an unnecessary content to make the manuscript clearer as well as added an information about herpes simplex virus. Please see details in Section 3.2 and Section 6.3.2.  In addition, we have deleted a part of contents about feline coronavirus but remained the relationship between feline coronavirus and useful inhibitor GS-441524, in the Section Introduction. Besides, we adjusted structure of the manuscript in order to make it more logically. Please see details in Section 3.2 and Section 3.3.

4、The Authors should decide if they focus on natural antiviral products or rather they compare natural products with antiviral drugs. In the manuscript the description of drugs dominates, and only a few natural compounds were mentioned. The title and the abstract should correspond to the main text but in this version of the manuscript it is not.

Response: We appreciate the reviewer's comment. We have revised Abstract to make a match of the main text. The novel antiviral agents designed based on protein structure are almost chemicals, that means expensive expense and toxicity. In this paper, we summarized the chemical agents in order to provide the antiviral targets for screening cheaper and safer natural products with potent antiviral activity similar to the chemicals. Please see updated discussion.

5、The references in the text should be improved, sometimes author’s name and year are included instead number.

Response: We have corrected the references, please see the updated manuscript.

6、Also, unnecessary abbreviations were in the text, for example, psoromic acid PA) – this term is the only one in the text.

Response: We have checked all of abbreviations in the text, such as deleted the abbreviation of poromic acid.

Reviewer 2 Report

Herpes simplex virus should be used instead of just herpes virus.  Do not use the abbreviation HSV without first using the term, followed by the abbreviation.

Section 2.1. Coronaviruses are positive -sense

Section 3.2  It would be clearer for line 2 to include “each” HSV 5

Section 3.3  Reports on use of chloroquine not balanced.  Did not include many recent studies that demonstrate it is not effective and show no benefit as antiviral.

Section 4

Line 1 All viruses use the host cell translation system – not just RNA viruses.

Section 4.1.2. Last paragraph – first line implies that viruses use their own translation system.  Clearly state that viruses depend on cellular translation system.

Section 5  - use the words “Toll-like receptor” (TLR) before using the abbreviation

Section 6.1 – no citations

Section 6.1.1 Last full sentence states that researchers had found some natural products through molecular docking but authors do not cite any of these.

Section 6.2.2  Research on tea catechins is extensive but researchers only cite one study

Section 6.3.3 Again, authors used the general term Herpes virus  to refer to which of the human herpes viruses.  I assume authors are referring to Herpes simplex virus.

Some sections include a summary and others do not.  Why?

Discussion:  Why wasn’t glycyrrhizinic acid study reviewed in the section on coronaviruses?

Format of citations inconsistent in discussion (superscript used – 5 and 6)

Formatting of References is inconsistent (1 - 50 used one format and 51 - 115 used a different format. 

There are no figure legends.

Author Response

Manuscript ID: viruses-2303505

Title: Broad-spectrum antivirals derived from natural products

Responses to reviewer2

Reply: Thank you for your constructive comments and suggestions. Considering the suggestions, we have conducted a thorough revision to make the study clearer and stronger. All changes are highlighted in blue in the revised version, a "Marked Up Manuscript File," and responses to the reviewers' points are addressed in detail below.

1、Herpes simplex virus should be used instead of just herpes virus.  Do not use the abbreviation HSV without first using the term, followed by the abbreviation.

Response: We appreciate the reviewer's comment. The paper has been thoroughly revised.

2、Section 2.1. Coronaviruses are positive -sense

Response: It has been corrected. Please see Section 2.1.

3、Section 3.2 It would be clearer for line 2 to include “each” HSV 5  

Response: We deleted the description “5 glycoproteins” and changed the sentence to “HSV entry is quite different from that of an RNA virus. As we mentioned above, there are different receptors for HSV, which means that the inhibitors targeting HSV receptors may be vulnerable to compensatory action.”

4、Section 3.3 Reports on use of chloroquine not balanced.  Did not include many recent studies that demonstrate it is not effective and show no benefit as antiviral.

Response: We have revised the part of chloroquine and reconsidered recent reports. Please see page 4, and lines 40-42(unmarked version).

5、Section 4 Line 1 All viruses use the host cell translation system – not just RNA viruses.

Response: We have revised it. Please see in Section 4 in which Both DAN and RNA viruses utilize host cellular translation system.

6、Section 4.1.2. Last paragraph – first line implies that viruses use their own translation system.  Clearly state that viruses depend on cellular translation system.

Response: In the original text “the viral translation system” does create ambiguity. We have changed it into “Since viruses and cells use common translation system, on which inhibitors targeting are difficult to apply in a clinical setting, due to their cytotoxicity.”

7、Section 5  - use the words “Toll-like receptor” (TLR) before using the abbreviation

Response: We appreciate the reviewer's comment. It has been updated.

8、Section 6.1 – no citations

Response: We have updated the citations in this section, please see Section 6.1.

9、Section 6.1.1 Last full sentence states that researchers had found some natural products through molecular docking but authors do not cite any of these.

Response: We have cited 4 compounds targeting on viral protein Nsp1. They are screened by molecular docking. Please see page 7, and lines 3-6 (unmarked version).

10、Section 6.2.2 Research on tea catechins is extensive but researchers only cite one study

Response: We have added some references about the antiviral effects of tea catechins. Please see page 7, and lines 15-16 (unmarked version).

11、Section 6.3.3 Again, authors used the general term Herpes virus to refer to which of the human herpes viruses.  I assume authors are referring to Herpes simplex virus.

Response: We appreciate the reviewer's comment. The paper has been thoroughly revised.

12、Some sections include a summary and others do not.  Why?

Response: The paper has been thoroughly revised. We adjusted, canceled, or added a summary in different parts of manuscript. Please see page 9, and lines 1-11 (unmarked version).

13、Discussion:  Why wasn’t glycyrrhizinic acid study reviewed in the section on coronaviruses?

Response: We found that the mechanism of action of glycyrrhizic acid is complicated, and it is difficult to fit it into a chapter of this paper. We listed the research progresses in this natural product in the Section Discussion, based on the references indicated below.

  1. Rai H, Barik A, Singh YP, Suresh A, Singh L, Singh G, Nayak UY, Dubey VK, Modi G. Molecular docking, binding mode analysis, molecular dynamics, and prediction of ADMET/toxicity properties of selective potential antiviral agents against SARS-CoV-2 main protease: an effort toward drug repurposing to combat COVID-19. Mol Divers. 2021 Aug;25(3):1905-1927. doi: 10.1007/s11030-021-10188-5. 
  2. Luo P, Liu D, Li J. Pharmacological perspective: glycyrrhizin may be an efficacious therapeutic agent for COVID-19. Int J Antimicrob Agents. 2020 Jun;55(6):105995. doi: 10.1016/j.ijantimicag.2020.105995.

14、Format of citations inconsistent in discussion (superscript used – 5 and 6)

Response: We appreciate the reviewer's comment. It has been updated.

15、Formatting of References is inconsistent (1 - 50 used one format and 51 - 115 used a different format. 

Response: We appreciate the reviewer's comment. It has been updated.

16、There are no figure legends.

Response: The figure legends were not involved in the previous version of manuscript, and we have attached it to the end of the article.

Round 2

Reviewer 1 Report

I don't have any comment. I accept the amended version of manuscript.

Author Response

Manuscript ID: viruses-2303505

Title: Broad-spectrum antivirals derived from natural products

Reply: The authors would like to thank you for constructive comments and suggestions. Considering the reviewer’s suggestions, we have corrected our mistakes to make the text clearer. We greatly appreciate the valuable guidance provided by the reviewer in improving the clarity and flow of our text. Thank you once again for your contribution to our work.

All changes are highlighted in blue in the revised version, a "Marked Up Manuscript File," and responses to the reviewer’s points are addressed in detail below.

  1. Abstract “which means the less resistance” should be which means less resistance.

Answer:We appreciate the reviewer's comment. We have corrected this sentence. Please see in 1, lines 11.

  1. Very awkward sentence; not clear. Should re-state the following: “With the recent revelation of bearing in mind virus replication mechanisms, however, and the development of molecular docking technology, …”

 Answer:We appreciate the reviewer's comment. We have changed this sentence into “New techniques and ideas are currently being developed for the design and screening of antiviral drugs, thanks to recent revelations about virus replication mechanisms and the advancement of molecular docking technology.” Please see in 1, lines 12-15.

  1. p 2  Line 16  Remove “The “ preceding HSV

Answer:We appreciate the reviewer's comment. We have corrected this sentence. Please see in page 2 line 9.

  1. p 3  Line 11 Spelling error - should be mediated not meditated

Answer:We appreciate the reviewer's comment. We have corrected this spelling. please see in page 3 lines 3-4.

  1. p 4 Line 27  Authors should use herpes viruses not HSV because they are referring to a property of several different herpes viruses, not just HSV.

Answer:We appreciate the reviewer's comment. We have We have changed all of HSV in this paragraph to herpes virus. Please see in page 4, lines 10-15.

  1. p 5 Line 2 “CQ is s useless” – remove s

Answer:We appreciate the reviewer's comment. We have corrected this sentence. Please see in page 4 line 40.

  1. p 5 Line 2 The following statement. “So that more clinical data is required before widely treatment”.  This statement must be re-stated.

Answer:We appreciate the reviewer's comment. We have corrected this sentence.

  1. p 5 Line 11  “All of viruses” – remove of

Answer:We appreciate the reviewer's comment. We have corrected this sentence. Please see in page 4, line 49.

  1. p 5  Lines 40-41 “inhibitors targeting…are difficult”. Need to insert the translation system or something comparable.

Answer:We appreciate the reviewer's comment. We have changed this sentence into “Since viruses and cells use common translation system, on which inhibitors targeting this system are difficult to apply in a clinical setting, due to their cytotoxicity.” Please see in page 5, lines 26-27.

  1. p 7 Line 1  I suggest restating to: CoV encodes five structural proteins.

Answer:We appreciate the reviewer's comment. We have corrected this sentence. Please see in page 6, line 28-29.

  1. p 7 Line   33  “maybe” should be may be

Answer:We appreciate the reviewer's comment. We have corrected this usage. Please see in page 7, line 3.

  1. p 7 Line 34 – Usage error; I suggest replacing “besides” with Additionally

Answer:We appreciate the reviewer's comment. We have corrected this usage. Please see in page 7, line 3.

  1. p 7 Line 35. Usage error; “show the potential inhibitory effect”; should replace the definite article “the” with  “a”

Answer:We appreciate the reviewer's comment. We have corrected this usage. Please see in page 7, line 5.

  1. p 7 Line 36:  I suggest inserting the word “against” before SARS-CoV-2.

Answer:We appreciate the reviewer's comment. We have corrected this usage. Please see in page 7, line 5.

  1. p 8 Line 50. Usage error. RNA virus “genome” RNA should be RNA virus genomic RNA

Answer:We appreciate the reviewer's comment. We have corrected this usage. Please see in page 7, line 17.

  1. p 9 Line 18 I suggest deleting “As we mentioned above,”

Answer:We appreciate the reviewer's comment. We have corrected this usage. Please see in page 8, line 35.

  1. p 9 Lines 44-45.  “So that antiviral agents should be screened and used specificity to make them more effectively.”  This sentence does not make sense – I’m not sure how to correct this.

Answer:We appreciate the reviewer's comment. We have deleted this sentence.

Reviewer 2 Report

Abstract “which means the less resistance”  should be which means less resistance.

Very awkward sentence; not clear. Should re-state the following: “With the recent revelation of bearing in mind virus replication mechanisms, however, and the development of molecular docking technology, …”

p 2  Line 16  Remove “The “ preceding HSV

p 3  Line 11 Spelling error - should be mediated not meditated

p 4 Line 27  Authors should use herpes viruses not HSV because they are referring to a property of several different herpes viruses, not just HSV.

p 5 Line 2 “CQ is s useless” – remove s

p 5 Line 2 The following statement. “So that more clinical data is required before widely treatment”.  This statement must be re-stated.

p 5 Line 11  “All of viruses” – remove of

p 5  Lines 40-41 “inhibitors targeting…are difficult”. Need to insert the translation system or something comparable.

p 7 Line 1  I suggest restating to:  CoV encodes five structural proteins.

p 7 Line   33  “maybe” should be may be

p 7 Line 34 – Usage error; I suggest replacing “besides” with Additionally

p 7 Line 35. Usage error; “show the potential inhibitory effect”; should replace the definite article “the” with  “a”

p 7 Line 36:  I suggest inserting the word “against” before SARS-CoV-2.

p 8 Line 50. Usage error. RNA virus “genome” RNA should be RNA virus genomic RNA

p 9 Line 18 I suggest deleting “As we mentioned above,”

p 9 Lines 44-45.  “So that antiviral agents should be screened and used specificity to make them more effectively.”  This sentence does not make sense – I’m not sure how to correct this.

Author Response

(The authors gave the same response as above.)
